# HIV Status Disclosure to Adolescents Who Are Perinatally Infected in Rustenburg Sub District Northwest Province

**DOI:** 10.3390/children9121989

**Published:** 2022-12-17

**Authors:** Happy Maybe Maambiwa Khangale, Ndidzulafhi Selina Raliphaswa, Azwidihwi Rose Tshililo

**Affiliations:** Department of Advanced Nursing Science, University of Venda, Private Bag X 5050, Thohoyandou 0950, South Africa

**Keywords:** adherence, adolescents, antiretroviral therapy, disclosure, perinatally, parents, caregivers

## Abstract

Acquired Immune Deficiency Syndrome (AIDS) is a viral disease caused by Human Immunodeficiency Virus (HIV) which affects the immune system of human body. This study sought to explore how adolescents with perinatal HIV infection learn about their status as well as investigate their preferences about the disclosure process. A qualitative exploratory, descriptive, and contextual research design was used to explore the disclosure of an HIV-positive status among adolescents on antiretroviral therapy. Nonprobability purposive sampling was used to select the healthcare facilities, and adolescents were chosen using convenience sampling. In-depth individual interviews were used to collect data from the participants until data saturation was reached. Collected data were analysed using Tesch’s eight steps. The results of the study revealed that adolescents have been on ART (antiretroviral therapy) without the knowledge of their own status but taking ART. Delayed disclosure of an HIV-positive status to adolescents lead to adolescent not adhering to ART and wondering why they are on treatment while other adolescents are not. The study further revealed that parents and caregivers struggle to disclose an HIV-positive status of their children, leading them to lie about what the treatment is for, for example, that it is a treatment for cough.

## 1. Introduction

A growing percentage of HIV-infected children are surviving into adolescence because of the expansion of antiretroviral therapy (ART). According to the World Health Organization (WHO) and the joint United Nations programme on HIV/AIDS (UNAIDS), in 2020, there were 37.7 million [30.2–45.1 million] people living with HIV, including 36.0 million [28.9–43.2 million] adults and 1.7 million [1.2–2.2 million] children (0–14 years). In addition, 53% of all people living with HIV were women and girls. Of all people living with HIV, 84% [67–98%] knew their HIV status in 2020, and about 6.1 million [4.9–7.3 million] people did not know that they were living with HIV in 2020.

One of the elements linked to better adherence is the disclosure of HIV status. However, the prevalence of disclosure in children and adolescents varies depending on the environment and the age of the patients, ranging from 13 to 60% in children between the ages of 5 and 17 from Asia or Southern-Eastern Africa [1].

For adolescents, confidentiality may be jeopardized by disclosure because it may exacerbate mental and behavioural issues, familial disputes, or perceptions of societal stigma [2].

A recent study from Zimbabwe showed that learning about their HIV status is still one of the most difficult life events for adolescents living with HIV/AIDS [3]. Despite these facts, the World Health Organization’s (WHO) guidelines for HIV status disclosure are limited to children under the age of 12 [4], even though many perinatally-infected children are not disclosed of their status until they become adolescents [5]. Healthcare workers and caregivers have minimally-tailored guidance on how to approach the issue of disclosure to these adolescents, other than that full disclosure is encouraged and should occur in “developmentally appropriate” stages [6]. The WHO recently developed new guidelines for HIV testing and counselling in adolescents [7], however, these guidelines do not address the issue of disclosure to adolescents, and only deal with the disclosure of adolescents’ HIV statuses to others [8]. For disclosure to adolescents, the new guidelines simply defer to the above-mentioned existing guidelines for children under 12 [9].

The delayed disclosure of a child’s HIV-positive status during the transition to adolescence affects their adherence to treatment, causing a high number of unsuppressed adolescents. Hence, we assessed and explored the knowledge of adolescents’ own HIV-positive status.

## 2. Materials and Methods

### 2.1. Study Design

The study adopted a qualitative exploratory, descriptive, and contextual design to explore the disclosure of an HIV-positive status among perinatally-infected adolescents on ART. The design was chosen as the participants can freely narrate their experiences of being on ART treatment without the knowledge of what the treatment is about.

### 2.2. Data Collection

The study was conducted utilizing a qualitative research approach, which means that the qualitative data-gathering method was used to collect information from the participants. The study was conducted in one of the South African Provinces in the Northwest. A semi-structured interview was utilized as one of the qualitative data-collection techniques. The interviewer asked two questions followed by additional probing, and both close-ended and open-ended questions were included in the semi-structured interview in Appendix A.

Adolescents were recruited to participate in the study during their follow-up visit to the healthcare facilities in the Rustenburg sub-district of the Northwest Province. A pretest was conducted before the actual study began to detect and correct any flaws that might occur in the actual study. However, participants who were part of the pre-test were not included in the main study. The researcher scheduled appointments with the guardians of all participants under the age of 18 so that they could sign the consent form for their children to participate in the study. After receiving complete information regarding the research study (the benefits and risks), informed consent was also obtained from participants aged 18 and 19 years. This was completed after the adolescents on antiretroviral therapy between the ages of 15 and 19 years were informed about the study. Participants were also informed that participation was voluntary, and that they were able to withdraw from the study at any time.

The participants who consented to participate were asked to give their consent for a tape recorder to be used to record the interviews to capture all the information provided by the participants. The researchers purposively collected data from the sampled adolescents. Only 14 participants were included in the study as data was saturated at the 10th participant. However, interviews were continued until the 14th participant. This was completed to ensure that the data had indeed saturated, as no new information was provided. The research interview with the participants took place in their respective homes and in the research facilities. Each interview lasted for at least 30–45 min. All measures of trustworthiness were ensured including credibility, dependability, transferability, and conformability. Member check was also completed by going back to the participants with captured data so that they could confirm that the information that had been captured was the true reflection of what they said. All measures to reduce the spread of COVID-19 were adhered to throughout the study including hand washing, sanitizing, maintaining a social distance of 1.5 m, and masking.

### 2.3. Ethical Considerations

Approval to conduct the study was obtained from the University of Venda ethical clearance committee SHS/20/PDC/40/2110 and the approval date is 23 October 2020. The Department of Health, Northwest Province also gave approval to access the selected health facilities.

### 2.4. Data Analysis

In this study, individual verbatim comments from tape recordings were played and transcribed, and those delivered in Tswana were translated into English. Each transcription was examined separately before being discussed with the supervisors, as well as the final arrangement of the themes and sub-themes. Tesch’s eight steps of the open coding method were used to analyse the data in this study. As stated by Lincoln and Guba (1985) and Krefting (1991), researchers used different techniques to ensure data trustworthiness, including credibility, dependability, conformability, and transferability. The researcher spent time with each adolescent to build trust. Purposeful probing was used for member checking throughout the interview. The researcher’s findings were discussed with the participants. For dependability, experts were used to validate the methodology, which was further enhanced using an independent coder to ensure consistency. To ensure conformity, notes were retained in a secure location to allow for the creation of an acceptable trail and the determination of conclusions, interpretations, and suggestions if their sources could be tracked.

## 3. Results

### 3.1. Biographical Information of Participants

The study comprised 14 participants, all of whom were adolescents, who were willing to participate and met the criteria as indicated in Table 1 below. Of the 14 participants, 5 were less than 18 years old and 9 were 18 to 19 years old. The majority of participants (11) were on ART for more than 10 years whereas 3 were on ART for more than 2 years. Furthermore, 12 participants were still in secondary school whereas 2 were in postsecondary school.

The study findings revealed one major theme accompanied by five sub-themes as indicated in Table 2 above. The major theme is knowledge related to HIV+ status and related factors, lack of versus the existence of knowledge related to the importance of adherence to treatment, lack of versus the existence of knowledge related to the initial diagnosis, side effects, lack of knowledge related to the HIV+ status and how it was transmitted, and disclosure of HIV-positive status viewed as important.

### 3.2. Theme 1: Knowledge Related to HIV+ Status and Related Factors

Theme 1 is divided into five sub-themes which describe the knowledge related to HIV+ status and related factors.

#### 3.2.1. Sub-Theme 1.1: Lack of Versus the Existence of Knowledge Related to the Disease Condition from Childhood to Date Explained

Participants in this study revealed that they did not have knowledge related to their HIV status and other factors related to it. This was supported by participants who expressed themselves as follows:

Participant 1 said, “(Looking down angrily) I found out a few months ago when I went to get my prescription with my mother, and she took me to the counselling room with her, where I was told everything, I never thought I might be HIV positive as my mother didn’t tell me the truth about the medication that I have been using every day. Telling me the truth while we were at home could have at least been something to me (crying)”.

One participant only learned about her own perinatally-acquired HIV-positive status after being sick and visiting a local clinic, where her status was revealed to her.

Participant 7 said, “I was sick and came to the clinic and tested positive for HIV and when I arrive home my mother was not surprised because she knew I was born with it. I couldn’t even look at her as I was disappointed in her for not taking care of me, for not disclosing to me about my status and make sure I take treatment. I had many questions as to why she could let me be sick while she knows the truth behind my sickness. Learning about ones HIV positive status and find only found out that you were born with it felt like a betrayal to me by my mother”.

#### 3.2.2. Sub-Theme 1.2: Lack of Versus the Existence of Knowledge Related to the Importance of Adherence to Treatment and Its Future Implications Described

Participants specified that they could see the benefits of adhering to ART, as they were healthy and stronger with treatment. Participants responded in the following ways:

Participant 12 said, “Yes, I see the benefits because when I’m on treatment, people don’t realize I’m sick because my body is healthy. Taking treatment have always made my blood results to me good when I am checked. The nurse’s health education on treatment importance played a major role for me to know that I shouldn’t skip treatment doses. Treatment prevented me from being sick by other diseases”.

Participant 10 said, “Yes, I see the benefits because treatment has made me stronger from birth. The treatment is the one which have enabled me to transition from childhood to adulthood. It made me to still look like other adolescents although I am HIV positive. The treatment increases lifespan”.

The participant further indicated that although there are benefits of treatment, he also experiences side effects which he does not regard as a benefit. This is how he expressed himself:

“Yes, there are benefits; for example, you don’t get sick as much when you’re on treatment (looking down); but there are times where there are no benefits because I sleep and don’t finish my schoolwork. Taking treatment early results in me falling early that’s why I hardly complete schoolwork. It becomes tough because my mother does not allow me to change times as we were told at the clinic that treatment is taken only in one time so that it can work for me”.

Participant 5 said, “I see the benefits, but I have my doubts because I am receiving treatment and instead of studying, I am sleeping, therefore I am not benefiting. Sleeping result in me not performing well at school which is not good for my future”.

Participant 8 said, “Yes, there are additional benefits to receiving treatment, such as being protected from becoming ill and making it difficult for others to notice that you are HIV positive. I was told by the nurse that when I don’t take treatment everyone will know that I am sick. My treatment makes my blood to be strong. I have faith in life because of my treatment”.

The benefits of recovering from opportunistic infections by being on treatment was also indicated by the participant.

Participant 3 said, “Yes, I see more benefits because I was sick and ended up with TB when I wasn’t taking treatment, and I’ve seen the benefits of taking treatment after starting treatment. The time I was sick not taking treatment I had given up in life with no hope, but the treatment brought life and hope to me as I have been well since taking it. I have indeed experienced it that ART increases one’s lifespan, I wouldn’t be living today if it was not for the treatment. It made my cd4 cells to be strong”.

#### 3.2.3. Sub-Them 1.3: Lack of Versus the Existence of Knowledge Related to the Initial Diagnosis, Side Effects, Together with the Signs and Symptoms Explained

The study revealed that adolescents lack knowledge related to the initial diagnosis of HIV. The side effects together with the signs and symptoms after the initial diagnosis were explained by different participants who emphasized the difficulty of engaging in their daily routines after taking treatment. The following quotes were made:

Participant 4 said, “I was struggling to cope with the side effects after taking treatment since I couldn’t do anything but sleep, especially schoolwork, which I couldn’t do after taking treatment, but now that the school day has changed, I only have trouble waking up and feeling sleepy in the mornings”.

Participant 3 said, “I’ve also failed in school since I’ve been unwell and missed school days, this was because of the delay of taking treatment which made me to have opportunistic infections that required me to be hospitalized. When I started treatment, I had no energy to study after taking treatment all I want was to sleep. I experienced the feeling of nausea and dizziness the first week of starting treatment”.

Participant 13 said, “As for me, I’ve noticed that being positive might make you feel different from other children, particularly in the first months following disclosure. You always feel like you no longer worthy to be amongst others, what if they find out? Will they accept my condition. The fear of losing my loved ones in the community due to my status took over in the first month of disclosure. The dizziness I have after treatment forces me to go to bed early, sometimes I don’t manage to do all my schoolwork”.

In this study, participants had difficulties coping with the side effects during the first month of diagnosis. Therefore, the researchers observed that support for adolescents during their first month of treatment is important for them to endure the side effects and regain hope after being diagnosed with a lifelong condition.

#### 3.2.4. Sub-Theme 1.4: Lack of Knowledge Related to the HIV+ Status and How It Was Transmitted (Parents Blamed)

The participants in the study indicated that they lack knowledge about their HIV+ status. Adolescents took treatment without the knowledge of their status, which led to them to put more blame on their parents after disclosure of their status and the way in which HIV was transmitted to them. Participants who were infected through mother-to-child transmission believed that their mothers would have taken treatment to prevent mother-to-child transmission.

Participant 6 said, “Yes, she did after a bit of coercion. My mother had been on ART for years and informed me when she was pregnant that she wasn’t taking her treatment properly, which resulted in her giving birth to a positive baby, which was me, and she never made me to take treatment; instead, she remained silent and waited for me to find out on my own. It was never easy to me to accept because she could have prevented me from being infected by adhering to treatment while she was pregnant. I felt like my mother betrayed me and didn’t love me enough she could have at least taken treatment to give birth to me without HIV positive (angry)”.

While some participants explained how HIV was transmitted to them, other participant had the questions regarding why they were born negative, but tested positive for HIV after a few months of breastfeeding.

Participant 10 said, “Yes, she did inform me that I was born HIV negative, but that I tested positive for HIV/AIDS within six months of breastfeeding. It made it difficult for me to comprehend the process of becoming positive while breastfeeding. I had many questions as how come and my mother told me that her blood was not well which made me to be infected. My mother should have at least not breastfed. Her blood was not good because she didn’t adhere to treatment”.

Participant 9 said, “My mother told me when I was 15 years old that she was diagnosed with HIV as she was about to give birth in her last month, which caused her to give birth to an HIV-positive child, of which I am the child. She said she was not using condom as she was pregnant which made her to be infected with HIV. They tried to give her treatment after being diagnosed but it was late to save me from getting HIV”.

Adolescents with perinatal HIV were dissatisfied with how they contracted the virus and blamed their parents for not doing more to prevent them from contracting it.

#### 3.2.5. Sub-Theme 1.5: Disclosure of HIV-Positive Status Viewed as Important

Participants in the study thought it was crucial to disclose their HIV-positive status because it affects adherence to their treatment. Knowing why they need to take treatment motivates them more than not knowing. Participants in the study shared their experiences before and after disclosure, emphasizing the importance of disclosure. This is supported by the following quotes from the participants:

Participant 2 said, “Yes, it was difficult because I had no idea it could be HIV treatment, and as a child, I had to be furious that I was not told and that I was constantly forced to take treatment that I had no idea what it was for. I was angry towards my parents for not telling me what the treatment was for. I felt like I was betrayed by not being told early that I am HIV positive. To be told you are HIV-positive at my age it was really hurting and not easy to accept as I had many thoughts as to what my friends will say if they find out about my status (looking down). I thought I wouldn’t make it face the world”.

The process of disclosure was delayed by parents and caregivers as they had to lie about their children’s status and not tell them the truth about the treatment. Poor disclosure to adolescents leads to poor adherence to treatment.

Participant 5 said, “I couldn’t take my treatment before disclosure because I didn’t understand why, and the treatment I was taking was so hot I couldn’t stand swallowing it, so I didn’t take it every day. Some days I pretended to take it when I didn’t, and after disclosure, I was able to take treatment knowing exactly what it was for. It was not easy to take treatment before disclosure because other children of my age were living freely. The disclosure helped me to adhere to treatment as I knew the danger of poor adherence. Although I was angry at my parents I for lying to me I continued taking treatment better than before”.

Another participant, despite being diagnosed with HIV during adolescence, still disclosed their status to family and received support rather than discrimination and stigma.

Participant 14 said, “I had support from my sister, who was able to sit with me and show me, love, by correcting me and telling me that no matter what had occurred, I should not lose hope and that there is still life even if I am HIV positive. I was afraid to disclose knowing how careless I was not protecting myself. My sister showed me love and support which gave me hope to adhere to treatment”.

## 4. Discussion

Participants in the study revealed a lack of understanding of their disease condition, as they only learned about it after visiting a healthcare facility because they were sick, when it was then revealed to them that they were HIV+. Moreover, other participants had been taking treatment without realizing it was ART, which was only revealed to them when they went for collection.

The study findings are supported by the study conducted by [10] which revealed that most of the individuals had been ill for a long time and had been hospitalized several times before receiving their ART treatment. A majority of the adolescents were still at school, therefore, their school activities were negatively affected because of their illness. Consequently, some of the adolescents had to repeat classes because they were unable to attend classes on a regular basis. Hence, studying was difficult for them due to their illness.

The study further revealed that adolescents had no knowledge regarding their HIV status until they developed opportunistic infection symptoms which led them to visit the clinics where they were diagnosed HIV+. However, knowing their HIV status helped them to be more positive about taking the treatment, even though it was not easy, since they underwent denial before accepting the situation. Knowing their HIV status allowed the participants to embrace the treatment’s importance in their lives, as to them, it appears that it is their reason for living—they have put their faith in receiving treatment because it keeps them healthy.

These findings are consistent with those of a study conducted by [10] which highlighted that an individual’s experiences determine adherence and non-adherence. Furthermore, the study revealed that an adolescent’s parents, caregivers, and healthcare providers influenced their treatment adherence by instilling in them a fear of dying. Therefore, the researchers observed from this study that adherence to treatment was viewed as important by most participants who believed that, if it was not for treatment, they would not be living. Contrary to the above finding, only one participant in the current study was perplexed by the importance of treatment, viewing it as significant when it did not allow him to complete his schoolwork, which was supposed to help him construct a future. Therefore, based on the above information, it is critical to emphasize the importance of health education on side effects to avoid noncompliance.

Furthermore, this study revealed that when adolescents are aware of their HIV status, they are better able to comprehend the importance of hospital visits and drug regimens, allowing them to make meaning out of their lives [11].

According to [12], adolescents living with HIV infection in Zambia reported going through changes after learning about their HIV-positive status, including enduring the obstacles associated with being infected with HIV/AIDS. After being diagnosed with a life-threatening illness, these children were able to survive. On the other hand, some of the children recounted feelings of despair after learning of their HIV-positive diagnosis, which came after a long time of illness. Consequently, the disclosure was both alarming and disempowering for these children, which required the need for counselling support. Regardless, caregivers also extended the revelation of their HIV-positive status which was influenced by mistrust, worries of stigma and discrimination, and secrecy [13]. Similarly, the findings of this study concurred with the findings of the study conducted by [14], who affirmed that participants displayed some anger and an attitude of blame toward their biological parents for infecting them with the virus. This is also consistent with the findings of the study by [15], who reported that children born with HIV who survive into adolescence are at risk of developing psychological disturbances because of long-term HIV and its related stressors, together with the long-term effects of medications.

For a variety of reasons, it is critical to inform children with HIV about their HIV status. The fact that they have been diagnosed with HIV is significant, and it is now part of their complete HIV care. According to some research, children who are told about their HIV infection are more likely to stick to their antiretroviral therapy (ART) [16], and [17] children who are disclosed of their status are four times more likely than nondisclosed children to adhere to ART. As a result, in both developing and developed countries, nondisclosure to children and adolescents with HIV has been highlighted as a barrier to adherence to ART [18]. Norms, taboos, and beliefs about talking about sex, a lack of family guidance by HCWs to disclose the HIV status to their children, and lack of family guidance by HCWs must all play a role in breaking the taboos and norms of caregivers of talking about sexuality and disclosure to their children to facilitate adherence to ART. To make an informed decision to take medication, adolescents must first comprehend the consequences of non-adherence on their health and well-being. The findings of this study are similar to those of [16], which discovered that disclosing HIV improves knowledge and understanding of the condition while also lowering risky behaviours. Unprotected sexual intercourse and unintentionally infecting a partner are two risky activities that adolescents engage in. When people are informed of their HIV status, they can take steps to protect their health and reduce the risk of infecting others [19].

## 5. Conclusions

This study revealed an essential issue regarding HIV status disclosure to adolescents. It is important for communities to be educated on the importance of continuous HIV testing, knowledge, and understanding of HIV and ART. This should include the signs and symptoms of HIV/AIDS together with opportunistic infections. The information aimed to assist all individuals in limiting their shock and confusion after being diagnosed with HIV. Parents also need to know the importance of disclosing their HIV status to their adolescents, before they find out the information on their own, to prevent blame and anger.

Therefore, each circumstance should be examined separately to make unique disclosure recommendations. Although many other factors were not included in this study and the disclosure experiences emphasized in this study were not substantial predictors of HIV status, disclosure to adolescents who are perinatally infected remains crucial. In the future, researchers should consider utilizing other research techniques to draw more thorough conclusions on the disclosure of HIV status to adolescents who are perinatally infected.

## 6. Recommendations

The medical and social support staff should conduct continuous counselling to promote early disclosure to parents raising children with perinatal-acquired HIV. Campaigns should also be conducted in communities to educate the public about HIV, reducing the stigma directed at people on ART. Future research should investigate the experiences of parents and guardians while disclosing the HIV-positive status of their children and the challenges they are facing. The department should strengthen support to parents during the process of disclosure of an HIV-positive status of adolescents.

## Figures and Tables

**Table 1 children-09-01989-t001:** Biographical information of the participants.

Participant Number	Age	Gender	Number of Years on Treatment	Level of Education	Perinatal Acquired HIV
1	18	F	16+	Grade 12	Perinatal
2	17	M	15+	Grade 12	Perinatal
3	18	F	2+	Grade 11	Perinatal
4	19	M	16+	Postsecondary	Perinatal
5	18	F	17+	Grade 12	Perinatal
6	18	M	10+	Grade 12	Perinatal
7	17	F	15+	Grade 11	Perinatal
8	18	M	17+	Grade 11	Perinatal
9	17	F	16+	Grade 11	Perinatal
10	17	F	15+	Grade 10	Perinatal
11	19	F	17+	Grade 12	Perinatal
12	17	F	16+	Grade 10	Perinatal
13	19	F	2	Postsecondary	Perinatal
14	18	M	12	Grade 12	Perinatal

**Table 2 children-09-01989-t002:** Themes and sub-themes that emerged from the study findings.

1. Knowledge related to HIV+ status and related factors	1.1 Lack of versus the existence of knowledge related to the disease condition from childhood to date explained 1.2 Lack of versus the existence of knowledge related to the importance of adherence to treatment and its future implications described 1.3 Lack of versus the existence of knowledge related to the initial diagnosis, side effects, signs, and symptoms explained 1.4 Lack of knowledge related to the HIV+ status and how it was transmitted (parents blamed) 1.5 Disclosure of HIV-positive status viewed as important

## Data Availability

Data is available on request.

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
