# Peer review of "HIV Status Disclosure to Adolescents Who Are Perinatally Infected in Rustenburg Sub District Northwest Province"

_children, 2022, doi:10.3390/children9121989_

Round 1
Reviewer 1 Report
This article addresses an important and unfortunately ongoing issue of delayed disclosure in children and adolescents with HIV, often due to parental experienced stigma and shame. I believe the study has the ability to contribute to the extant literature, but a few major and several minor issues need to be addressed as follows:
Major concerns:
1) Methods, lines 68-69: The following statement is problematic: “The researcher collected data from participants who were chosen based on the researcher's judgment. Adolescents who met the researcher's criteria were recruited to participate in the study.” These criteria need to be specified. As described automatically reduces the results to selection bias. The enrollment criteria need to be clearly defined in the methods section.
2) The description of the methods warrant greater detail and clarification of the required steps. As written, they are not reproducible. Perhaps including a table with the step-by-step process would be helpful.
3) Discussion, lines 194-199 – and before and after are confusing…As written, the Discussion reads more like an annotation of author [10], and not the reporting of discussing your own results. The Discussion section would greatly benefit from review and editing by an experienced author to aid in expected format and reading flow of your study’s findings.
Minor concerns:
1) Abstract, line 14: add “how” before learn
2) Abstract, line 15, recommend “disclosure process” instead of “revealing procedure.”
3) Abstract, line 20, data “were” – plural, rather than “was.”
4) Abstract, general comment – the abstract would benefit from review to improve grammar and reading flow. The introducation has better written quality.
5) Methods: First para: this paragraph is very redundant and would benefit from grammatical editing to improve clarify and reading flow.
6) Methods: lines 71-72: “The researcher scheduled appointments with all under-18-year-
old adolescent guardians so that they could sign the assent consent form for their children to participate in the study.” This needs to be clarified. Adolescents < 18 would sign assent for themselves, while the parent/legal guardian would sign consent. This statement would also be better served to follow the statement on lines 73-74 re: those 18 years or older.
7) Please add to your methods section that interviews were recorded. We don’t learn that until the data analysis section.
8) Results, line 92. Rather than “age 18 years and above,” recommend just stating “age 18-19 years”, since your sample was not older than 19.
9) Results: “tertiary” – given your global audience, it may be helpful to clarify what tertiary education means, either by stating “post-secondary” or “college,” something that helps clarify the term that is not universally understood.
10) Results: subthemes: a “,” (comma) is needed after every instance of “said” where participant quotes are provided. It was done correctly in one instance under subtheme 1.4.
11) Results, line 138 “roll out” prefer rewording to read “engage in”
12) Results, line 155 “participants who were perinatally infected” – All had perinatal transmission, so this wording is a little confusing. Please add a “,” (comma) after “Participants”, and again after “transmission” to clarify the clause/phrase as intended.
13) Results, line 167, please define PMTCT at first use.
14) Discussion, line 189: “study done by [10]” please include at least the first author’s name followed by et al. to aid the reader.
15) Discussion, lines 190-193 – are these your results, or are you still citing author [10]? If your results, this is the first we are learning about your sample missing school/repeating a grade as those results are not reported in the results section. This needs clarification.
16) Discussion: it would benefit the manuscript to provide a section about the barriers to disclosing HIV diagnosis to youth – why parents do not tell or delay telling their children.
17) Recommendations, line 241: “nursing staff” – please reword to “medical and social support staff” as the responsibility lies with the provider and entire treatment team.
18) Minor grammatical issues throughout that require careful review.
Author Response
Dear reviewer. Thank you so much for the time to review our manuscript. Your comments were valuable and assisted us a lot make our paper of quality
Reviewer 2 Report
“HIV Status Disclosure To Adolescents Who Are Perinatally Infected In Rusternburg Sub District Northwest Province is interesting topic, but unfortunately the content is not well written and presented, the paper needs major revisions.
Here are the main comments that should be considered__
Abstract reflects the whole manuscript. Do not repeat the sentence as authors written the same thing in introduction and methodology part, and nowhere mentioned results analysis, please also provide more condensed concise discussion. Complete the last sentence.
Line 12: Confusing statement, please rewrite it_
“The question of telling young kids they have HIV emerges when they mature and no longer face a direct risk of mortality”
Line 15-16: Please edit the sentence.
· At the first appearance in the abstract/introduction, abbreviations should be preceded by words for which they stand. These abbreviated forms should be used uniformly in the whole manuscript to maintain the consistency.
Line 26: Add more keywords. please think better and try to use words that you did not use on the title.
Line 33-37: This paragraph is too long and difficult to follow. The authors need to rephrase this paragraph, and perhaps break it into two or three sentences.
· Consent form (for below age 18) and interview questions (written), please provide just for journal record (it will not be published with your manuscript).
· If the participants have any comorbidities, please mention.
· Different background of readers will read your article, if possible, please mention about ART.
Line 85: “For dependability, experts 85 were used to validate the methodology, which was further enhanced using an independent coder to ensure consistency “ Please add more details of Question validation methodology.
· Result section is well presented, but few concerns. Why selected participants response added there in sub-theme, why not adding all responses.
· In discussion authors should critically discuss their results, not only provide the few published data, a more detailed exposition of the points for a deeper understanding would be desirable.
In discussion some information which is not appropriate in this section, when describing facts and comparing with the previous data information may be succinctly added to the discussion.
Additionally, I suggest the improvements as a summary with the following sections:
1. A relatively small number of participants included in this study, you should broad your study to have additional data.
2. Suggest improving the conclusions and future perspectives mentioning (venturing thoughts) which will be the possible clinical improvements by this futuristic approach.
3. There are a number of English language issues throughout the paper.
Author Response
Dear reviewer. Your time to review our manuscript is highly appreciated

Round 2
Reviewer 2 Report
Revision is rushed and incomplete, although authors revised the manuscript, and they rectified the sections as I quoted earlier, but they still need to critically discuss their results.
As suggested earlier, you should broad your study to have additional data.
Conclusion part even after revision does not explain well.
Manuscript requires a thorough correction by English language professional.
Author Response
Thanks once more for the time spent reviewing our manuscript
Revision is rushed and incomplete, although authors revised the manuscript, and they rectified the sections as I quoted earlier, but they still need to critically discuss their results.
Discussion part revised and critically discussed from line 302 to 344
As suggested earlier, you should broad your study to have additional data.
More data were added, and quotations beefed up 135 to 299
Conclusion part even after revision does not explain well.
Conclusion part revised to be more meaningful line 367 to 374
Manuscript requires a thorough correction by English language professional.
Editing done
Thanks
